# Anionic Polymerization Using Flow Microreactors

**DOI:** 10.3390/molecules24081532

**Published:** 2019-04-18

**Authors:** Yusuke Takahashi, Aiichiro Nagaki

**Affiliations:** Department of Synthetic Chemistry and Biological Chemistry, Graduate School of Engineering, Kyoto University, Nishikyo-ku, Kyoto 615-8510, Japan; takahashiy@sbchem.kyoto-u.ac.jp

**Keywords:** flow synthesis, anionic polymerization, styrene, alkyl methacrylate, block copolymer, end functionalized polymer

## Abstract

Flow microreactors are expected to make a revolutionary change in chemical synthesis involving various fields of polymer synthesis. In fact, extensive flow microreactor studies have opened up new possibilities in polymer chemistry including cationic polymerization, anionic polymerization, radical polymerization, coordination polymerization, polycondensation and ring-opening polymerization. This review provides an overview of flow microreactors in anionic polymerization and their various applications.

## 1. Introduction

A microreactor is a flow-type chemical reaction device, which typically consists of micromixers and microtube reactors at a micrometer scale. A number of significant chemical conversions have been conducted in flow microreactors and have received significant research interests from both academia and industry [1,2,3,4,5,6,7,8,9,10,11,12,13,14,15,16,17,18,19,20,21,22,23,24,25]. Recent studies identified unique characteristics of flow microreactors, which allowed many challenging, or even impossible, organic syntheses in macro-type batch reactors to be successfully performed. In a flow microreactor system, the selectivity of a chemical reaction can be dramatically improved by fast mixing and efficient heat transfer, which are derived from short diffusion paths and high surface-to-volume ratios, respectively [26,27,28,29]. Likewise, short residence time in a microchannel enabled extremely unstable intermediates to be used in numerous useful chemical reactions [30,31,32,33,34,35,36,37,38,39,40,41,42,43,44,45,46]. Moreover, the use of flow microreactor system is quite effective from the viewpoint of the improvement of some safety due to smaller reaction volume and higher surface to volume ratio compared to batch reactors. Collectively, these features of flow microreactor systems have opened up new possibilities for various chemical reactions for organic synthesis [47,48,49,50,51,52,53,54,55,56,57].

Applications of flow microreactors appear to be particularly promising in polymerization in which a series of repeated chemical reactions turns small monomers into chain-like polymers. The reviews on various polymerization methods performed in microreactors were published by Hessel et al., Wilms et al., Mcquade et al., Serra et al., Frey et al. and Junkers [58,59,60,61,62,63,64,65]. The main purpose of this review is to provide a brief up-to-date overview on applications of flow microreactors for anionic polymerization with the insight into industrial use; therefore, the paper focuses on synthetic research rather than theoretical research.

## 2. Characteristic Features of Flow Microreactors

Flow microreactors can influence the very essence of chemical reactions because of the characteristic features derived from having small flow-type structures.

(1) Fast mixing [66,67]: With highly active reagents, a reaction is initiated immediately after mixing solutions; therefore, it is crucial to reach solution equilibrium quickly to ensure proper control over such reaction. Mixing occurs due to molecular diffusion. Theoretically, the time required for molecular diffusion is proportional to the squared length of the diffusion path. Having a much shorter diffusion path, a microreactor can achieve a mixing rate which is not achievable in a macroreactor. The working principle of a typical micromixer (a multilamination-type micromixer [68]) is shown in Figure 1. The solution 1 and the solution 2 are divided into flow segments with small width by a microstructure. Efficient and fast mixing, owing to short diffusion paths, takes place at the interfaces of the flow segments.

(2) Temperature control: Heat is transferred between the interior and exterior of a reactor through the reactor surface; therefore, the surface area per unit volume of the reactor is a major factor that ensures excellent heat transfer. Volume is equal to the cubed length of a reactor and surface area is equal to the squared length: the shorter the length, the higher the surface-to-volume ratio. When compared micro spaces to macro spaces, the former generally have much larger surface-to-volume ratios (Figure 2). Hence, heat transfer occurs more rapidly in a flow microreactor, enabling fast cooling/heating and precise temperature control.

(3) Residence time control: The time duration during which a solution remains inside a reactor is called the residence time. In flow reactors, the residence time increases with the length of the channel and decreases with the flow speed. By shortening microchannel lengths, the residence time in flow microreactors can be greatly reduced. Extremely short residence time allows reactions involving unstable short-lived reactive intermediates to be precisely controlled. Unstable reactive species can be transferred to another location for the next reaction before they decompose (Figure 3). By taking advantage of this feature, chemical transformations that are very difficult or impossible in macroreactors can be achieved in microreactors [69,70,71,72].

## 3. Living Anionic Polymerization 

### 3.1. Controlled/living Anionic Polymerization of Vinyl Monomers

Living anionic polymerization has been one of the most important reactions for polymer synthesis since it was first reported by Michael Szwarc in the 1950s [73,74,75,76,77,78,79,80,81]. The nature of anionic polymer ends, being chemically active with no capping agent, makes this polymerization method especially powerful by utilizing these ends for various complex polymer syntheses such as end-functionalized polymers, block copolymers, star polymers and graft copolymers [82,83,84,85]. In addition, living anionic polymerization is considered to be more ideal than radical polymerization because the former, requiring no termination and chain transfer reaction, usually results in lower polydispersity index (PDI = Mw/Mn). However, the anionic polymerization has be performed under strictly dehydrated conditions since an anionic living polymer end is very sensitive to moisture. This requirement of reaction condition greatly hinders industrial application. In addition, cryogenic condition, such as −78 °C, is required when anionic polymerization is performed in polar solvents with batch macro reactors [86]. In terms of industrial application, the usefulness of the polymerization is deteriorated by such requirement. On the other hand, with nonpolar solvents, the polymerization can be carried out at higher temperatures but much longer reaction time is needed for completion. Despite the aforementioned restriction, a number of remarkable advancements have been made on polymer synthesis by living anionic polymerization performed in flow systems. In fact, the kinetic studies on anionic polymerization in a continuous flow have been reported by Szwarc and Schulz et al. [87,88] and Lochmann and Muller et al. [89,90]. This polymerization strategy continues to expand the possibility of living anionic polymerization using flow microreactor systems and to innovate a way of developing functional materials in the field of polymer chemistry.

### 3.2. Controlled/living Anionic Polymerization of Styrenes in Polar Solvent Using Flow Microreactor Systems

Anionic polymerization of styrenes is a very useful technique for polystyrene synthesis because the polymerization can be modulated to produce polymers with accurately targeted molecular weights (Mn) and molecular weight distributions (Mw/Mn). The polymerization is applied for the synthesis of structurally well-defined polymers such as end functionalized polymers and block copolymers [91]. With a polar solvent, batch microreactors should only be performed under cryogenic conditions such as −78 °C to control the reactivity. On the other hand, flow microreactors can be used under much milder conditions such as 0 °C as reported by Nagaki and Yoshida [92]. Solutions of styrene in tetrahydrofuran (THF) and *s*-BuLi (in hexane) were mixed in a T-shaped micromixer M and the polymerization occurred in a microtube reactor R. The reaction was quenched by MeOH at the reactor outlet, resulting in polystyrene with a narrow molecular weight distribution (Mn = 1200~20,000, Mw/Mn = 1.09~1.13) (Figure 4). Polymer synthesis of narrower molecular weight distribution would be achieved by further purification of reagents and/or their effective mixing, although no such results are reported.

Highly controlled polymer dispersity was due to two factors: precise control of the initiation reaction by the fast mixing of initiator and monomer and the speedy removal of reaction heat by precise temperature control. Moreover, the molecular weight was also easily controlled just by changing the flow rates of monomer and initiator solutions. Furthermore, this method can be applied for various styrene derivatives having silyl, methoxy, alkynyl and alkylthio groups on the benzene ring. Löwe and Frey also reported anionic polymerization of styrene at 20 °C using a flow microreactor [93]. Their method successfully produced polystyrenes having a broad range of molecular weight within several seconds while the molecular weight distributions were narrow (Mn = 1700~70,000, Mw/Mn = 1.09~1.41). In addition, the flow microreactor allowed the polymerization to be performed under more easily attainable conditions, reducing/eliminating the need for strict dryness of apparatus and high vacuum techniques in batch macro reactors. In a flow system, residual impurities and moisture can be removed just by running monomer and/or initiator solution through a reactor before a desired polymer product is collected at the reactor outlet. Löwe and Frey reported another flow microreactor application, anionic polymerization of 2-vinylpyridine (2VP) (Figure 5). When performed in batch macro reactors, the polymerization needs inorganic salts such as lithium chloride to control the reaction. However, the reaction can be quenched in a timely manner by precise residence time control so that it prevents carbanionic living polymer ends from reacting with electron-poor pyridine ring [94,95].

Syntheses of complex polymers can be performed by integrating several flow microreactors. In other words, a multistep reaction of structurally well-defined polymers such as end-functionalized polymers and block copolymers can be conducted in one go just by putting several micromixers and microtube reactors together while effectively utilizing the livingness of polymer ends. For example, polystyrenes with silyl group at their terminal were generated by functionalizing living polymer ends with chlorosilanes such as chlorotrimethylsilane and chlorodimethylvinylsilane [92]. The block copolymers composed of two different styrenes were also synthesized in quantitative yields at 0 and 24 °C in the integrated flow microreactor system (Figure 6). 

Another effective strategy for complex polymer synthesis is the functionalization of anionic living polymer ends with epoxides, which have high reactivity toward nucleophiles because of ring strain. Use of functionalized epoxides enables a further transformation. For example, the use of flow microreactors allowed various functionalized polymers to be readily generated by the following procedures: polymerization of styrene and end-functionalization with the various glycidyl ethers having acetal structures such as ethoxy ethyl glycidyl ether (EEGE), 1,2-isopropylidene glyceryl glycidyl ether (IGG) and 2-phenyl-1,3-dioxane glycidyl ether (PDGE) (Figure 7) [96]. The acetal and ketal protecting groups in the glycidyl ethers are stable toward the highly reactive carbanionic living polymer ends but they can be easily cleaved under acidic conditions to afford multi-hydroxyl end-functionalized polymers (Figure 8).

Living anionic polymerization has been also utilized for syntheses of complex branched polymers, including star polymers and dendrimer-like star-branched polymers. Having unique and interesting properties in solution and also in liquid and solid state, these branched polymers have attracted much attention from theoretical, synthetic and practical points of view. Among them, block copolymers having different polymer chains on a core are especially interesting. To synthesize such a structure, at the first step, only one polymer chain has to be selectively introduced to a core molecule, which has multiple reaction sites. In a batch macro reactor, an excess amount of multifunctional core is necessary to prevent the formation of multi adducts, requiring one extra step of separating unreacted core before proceeding to the next step [97,98,99]. A flow microreactor can effectively solve this problem by suppressing the disguised chemical selectivity [100,101,102,103,104,105,106,107,108,109,110,111]. As shown in Figure 9, the end-functionalization with one equivalent of dichlorodimethylsilane leads to the selective formation of a product that has a single polymer chain on silicon (Mn = 1400, Mw/Mn = 1.13) while the use of a batch macro reactor results in lower controllability (Mn = 1300, Mw/Mn = 1.21). Extremely fast 1:1 micromixing of the living polymer chain and dichlorodimethylsilane enables the selective introduction of a single polymer chain into silicon. Therefore, the subsequent reaction with another living polymer chain using an integrated flow microreactor system gives block copolymers having two different polymer chains on a silicon core. Based on the present method, more complex macromolecular structures such as miktoarm stars, star block copolymers and block graft copolymers would also be effectively synthesized.

### 3.3. Controlled/living Anionic Polymerization of Styrenes in Nonpolar Solvent Using Flow Microreactor Systems

Anionic polymerization of styrenes could be conducted in nonpolar solvents at room temperature in a macro batch reactor only when the reaction time was significantly prolonged. Another drawback of this reaction is that the polymerization needs to be performed with <20% by volume styrene because higher volumes of styrene leads to a rapid increase in reaction temperature. With flow microreactors, both disadvantages can be readily resolved. In fact, controlled anionic polymerization of styrene initiated by *s*-BuLi in cyclohexane as a nonpolar solvent can be conducted at 80 °C by using a flow microreactor system to obtain polystyrenes in quantitative yields within 1 ~ 5 min (Figure 10). In the case of the polymerization of styrene with higher monomer concentrations (25~42%, by volume styrene), the polystyrene can be generated at 60 °C in cyclohexane using an aluminum-polyimide microfluidic device (Figure 11) [112]. Moreover, the molecular weight distributions of the polymers are influence by channel patterns: straight, periodically pinched, obtuse zigzag and acute zigzag channels. 

### 3.4. Controlled/living Anionic Polymerization of Alkyl Methacrylates Using Flow Microreactor Systems

Poly(alkyl methacrylate)s with well-defined structures are of significant research interest as they are versatile materials—such as plastics, adhesives and elastomers— that contain a number of different reactive functions. Living anionic polymerization is considered a superior method for poly(alkyl methacrylate)s synthesis because it is much faster than living radical polymerization, requiring no capping agent. However, to obtain the polymers of narrow molecular weight distribution, cryogenic condition such as −78 °C is compulsory when using a batch macro reactor [113,114]. In industrial production, maintaining such low temperatures can be a huge environmental and economic burden, limiting its useful polymerization. Living anionic polymerization of alkyl methacrylates initiated by 1,1-diphenylhexyllithium using a flow microreactor produces the corresponding poly(alkyl methacrylate)s with controlled molecular weight distribution under more accessible temperatures (methyl methacrylate (MMA): Mw/Mn = 1.16, −28 °C), (butyl methacrylate (BuMA): Mw/Mn = 1.24, 0 °C), (*tert*-butyl methacrylate (*t*-BuMA): Mw/Mn = 1.12, 24 °C). Precise control of the reaction temperature and fast mixing of a monomer and an initiator seem to be responsible for successful polymerization (Figure 12) [115]. Moreover, polymethacrylates synthesized by flow microreactor polymerization might give a different structure compared with that of the conventional batch method because optimized reaction temperatures would be different.

For synthesis of end-functionalized polymers and block copolymers, it is crucial to control and maintain livingness of the reactive carbanionic polymer ends. The livingness of the polymer ends can be evaluated in a flow microreactor system as shown in Figure 13. A solution of an alkyl methacrylate and that of 1,1-diphenylhexyllithium are mixed in micromixer M1 and the polymerization is carried out in microtube reactor R1. Then, a solution of the same monomer is introduced at micromixer M2, which is connected to microtube reactor R2 where the sequential polymerization takes place. By changing the length of R1 with the fixed flow rate, how the residence time in R1 can influence the product can be analyzed. With any residence time, the addition of the second monomer solution resulted in molecular weight increase. However, the longer the residence time in R1, the molecular weight distribution also increases, presumably because of decomposition of the polymer end (Figure 14). By choosing an appropriate residence time in R1 (MMA: 2.95 s, BuMA: 0.825 s), the sequential polymerization can be successfully carried out without significant decomposition of the living polymer end [116]. Moreover, the subsequent reaction of the living polymer end with a different alkyl methacrylate leads to the formation of a block copolymer that has narrow molecular-weight distribution (Table 1).

### 3.5. Controlled/living Anionic Block Copolymerization of Styrenes and Alkyl Methacrylates Using Integrated Flow Microreactor Systems

As mentioned above, flow microreactors are effective for accomplishing the controlled anionic polymerization of styrenes as well as alkyl methacrylates. Characteristic features of flow microreactors, including fast mixing, fast heat transfer and precise residence time control, allows molecular weight distribution to be controlled in a precise manner even under more accessible conditions such as temperatures from 24 to −28 °C. Another advantage of flow-microreactor-controlled polymerization is the ease with which a reaction system can be expanded just by connecting multiple micromixers and microreactors, which can exclude extra steps of isolation and purification and requires minimal modulation to reaction conditions. In fact, by using integrated flow microreactor systems, the polystyrene living polymer end, which is produced by butyllithium initiated anionic polymerization of styrene, can be effectively trapped with 1,1-diphenylethylene and the resulting organolithium species can be used as a macro initiator for anionic polymerization of alkyl methacrylates. Therefore, styrene–alkyl methacrylate diblock copolymers can be synthesized with a high level of molecular weight distribution control at easily accessible temperatures such as from 24 to −28 °C (Figure 15) [117]. Moreover, triblock copolymers can be also synthesized by sequential introduction of styrene and two different alkyl methacrylates in a similar manner (styrene–*tert*-butyl methacrylate–methyl methacrylate triblock copolymer: Mn = 8800, Mw/Mn = 1.23, styrene–*tert*-butyl methacrylate–butyl methacrylate triblock copolymer: Mn = 9000, Mw/Mn = 1.35) (Figure 16).

The anionic polymerization of block copolymers comprising fluorine-containing monomers has been extensively studied due to the polymer’s unique and highly valuable physical properties; however, maintaining the livingness of their extremely unstable growth ends has always been too challenging, requiring an additive such as lithium chloride to control the reaction. Integrated flow microreactor systems proved to be effective to solve this problem, taking advantages of precise controls on temperature and residence time [118]. An example of one of the fluorine-containing monomers, 2-(nonafluorobutyl)ethyl methacrylate (NFBEMA), will be discussed. First, an optimal reaction condition is identified by analyzing the conversion rate of the first monomer and the molecular distribution of the block copolymer as shown in Figure 17 and Figure 18 (temperature: −40 °C, residence time: 7.9 s). Under the optimal condition, the NFBEMA polymer is generated and then is used for block copolymerization with alkyl methacrylates and alkyl acrylate, all of which can be performed in one flow (NFBEMA – *tert*-butyl methacrylate block copolymer: Mn = 29000, Mw/Mn = 1.15) (Figure 19). Moreover, the fluorine-containing triblock copolymer can also be synthesized by integrated flow microreactor system (NFBEMA–*tert*-butyl methacrylate–NFBEMA triblock copolymer: Mn = 16,000, Mw/Mn = 1.10).

## 4. Anionic Ring Opening Polymerization Using Flow Microreactor Systems

The ability of flow microreactors to effectively control highly reactive monomers can be utilized for reactions involving substantially dangerous compounds such as ethylene oxide [119]. Controlling reactions with ethylene oxide can often be quite difficult and the handling of the compounds requires a great deal of caution in batch macro reactors. In comparison, the distinctively small size of a flow microreactor can prevent reaction heat from accumulating and keep the amount of ethylene oxide used at any given time minimal, resulting in a much safer process [120]. The anionic ring opening polymerization [121] of ethylene oxide initiated by alkoxy anion is performed within 30 min residence time at 120 °C under pressurized condition, acquiring monoalkyl-ether terminated PEGs in high yield and narrow molecular weight distribution (Mn = 2100, Mw/Mn = 1.06) (Figure 20) [122].

The characteristics of flow microreactors, such as efficient mixing and fast heat/mass transfer, increase reaction rate. For organocatalyzed ring opening polymerization [123,124] of **ε**-caprolactone (CL) and **ε**-valerolactone (VL), the apparent polymerization rate constant (k_app_) in a flow microreactor is larger than that in a batch macroreactor (in flow microreactor: k_app_ = 0.00602 min^−^^1^ (CL) and 0.159 min^−^^1^ (VL), in batch reactor: 0.00286 min^−^^1^ (CL) and 0.079 min^−^^1^ (VL)) [125]. Furthermore, an integrated flow microreactor system enables block copolymerization even in ring opening polymerization (Figure 21). A series of monomer introduction produces well-defined poly(**ε**-valerolactone)-block-poly(**ε**-caprolactone) and poly(**ε**-caprolactone)-block-poly(**ε**-valerolactone).

## 5. Continuous Production Using Flow Microreactor Systems

One of the most important advantages of a flow microreactor is that a laboratory-scale reaction can be readily adapted to an industrial-scale production simply by continuously running a reaction for a long period of time. In a case of flow anionic polymerization of styrene initiated by *n*-BuLi, about 1 kg of polystyrene was produced over 3-h reaction time with high molecular weight and narrow molecular weight distribution (Mn = 8000, Mw/Mn = 1.1) [126]. The polymers of various molecular weights (Mn = 5000 to Mn = 14,000) with narrow molecular weight distribution (Mw/Mn = 1.08–1.16) were successfully synthesized simply by changing the relative flow rate and the concentration of solutions of styrene and initiator. In order to produce polymers with higher molecular weights, the suppression of pressure increase and/or the use of high pressure pump would be necessary.

In terms of the industrial production process, it is important to be able to stop and restart an operation easily. The Smoothflow pump has a structure which allows a solution to be contained in its pump head during a resting time. This feature offers significant superiority for a reaction operation with a reactant solution which is unstable to air and/or water. (Figure 22). A great deal of information has been acquired in laboratories and pilot plants, which should profit the further development of this field to realize commercial plants for making polymers in the near future [127].

## 6. Conclusions

The examples shown in this review demonstrate the great usefulness of flow microreactor systems for the structural control and production of polymers by anionic polymerization. Spatial integration improves the efficiency and speed of structurally-defined polymer synthesis by conducting a series of reactions in one flow. In addition, a reaction is controlled in a way that does not require extremely cryogenic conditions, removing one of the major obstacles for industrial use. Further development of the anionic polymerization method based on continuous flow synthesis is expected and this technology can greatly contribute to polymer production in the near future.

## Figures and Tables

**Figure 1 molecules-24-01532-f001:**
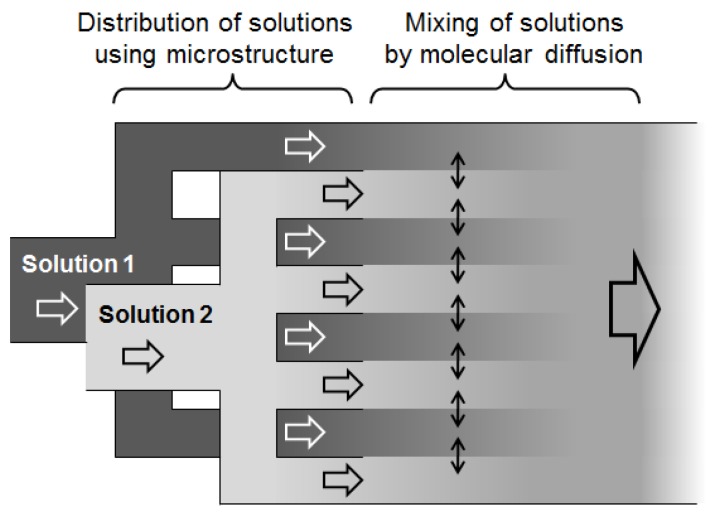
Working principles of a multilamination-type micromixer.

**Figure 2 molecules-24-01532-f002:**
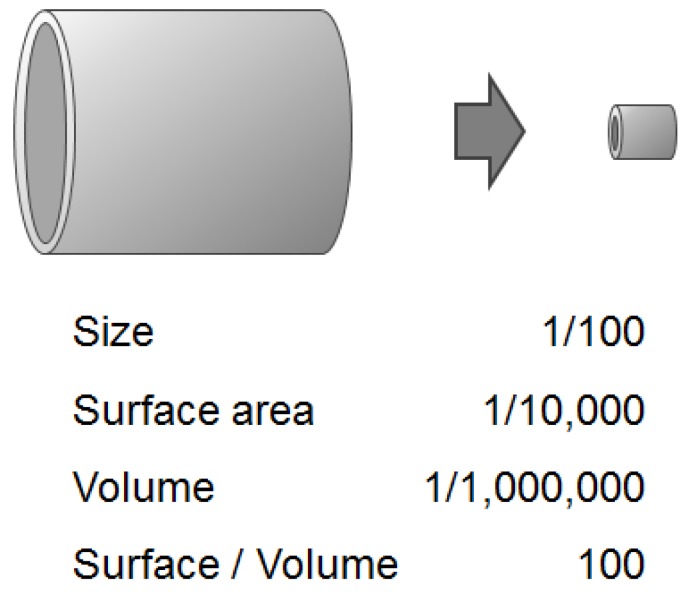
Numerical aspects of decreasing size. Effect of downsizing reactor on surface-to-volume ratio.

**Figure 3 molecules-24-01532-f003:**
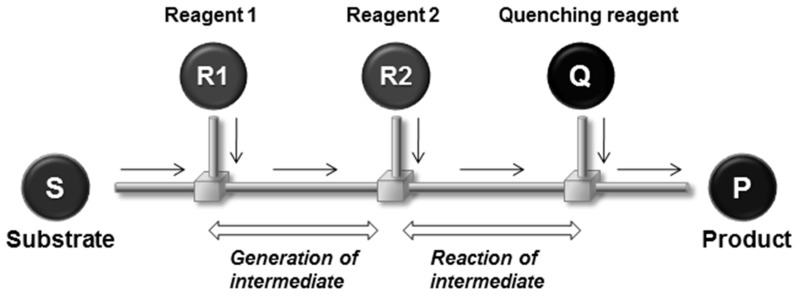
Principle of generation and reaction of unstable short-lived reactive intermediates based on residence time control in a flow microreactor.

**Figure 4 molecules-24-01532-f004:**
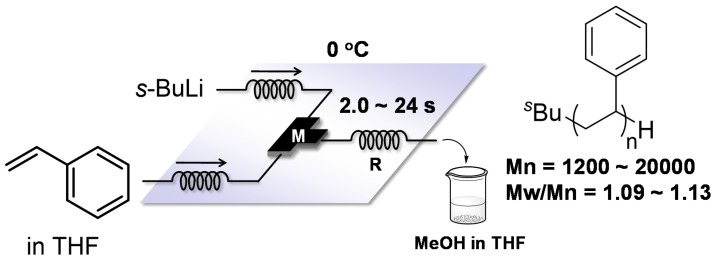
A flow microreactor system for anionic polymerization of styrene in tetrahydrofuran (THF). M: T-shaped micromixer; R: microtube reactor.

**Figure 5 molecules-24-01532-f005:**
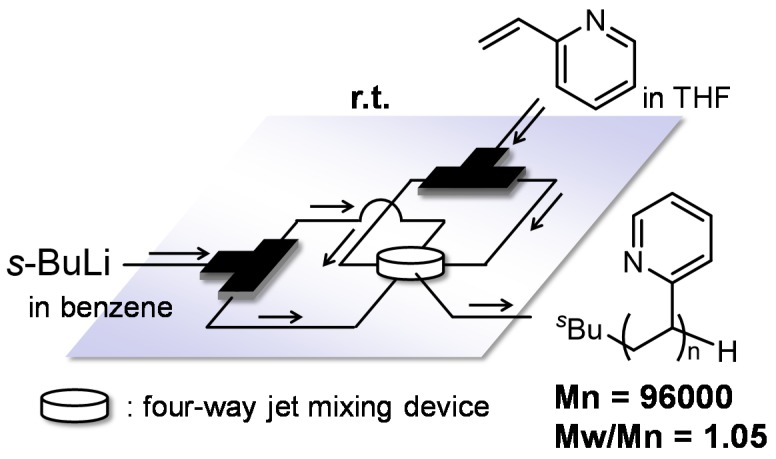
A flow microreactor system for anionic polymerization of 2-vinylpyridine in THF using four-way jet mixing device.

**Figure 6 molecules-24-01532-f006:**
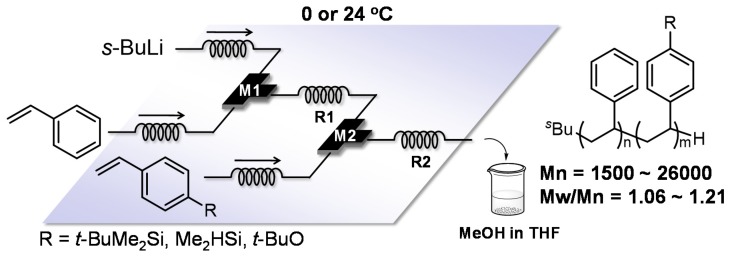
The integrated flow microreactor system for poly(styrene)-block-poly(styrene derivatives) synthesis in THF. M1, M2: micromixer; R1, R2: microtube reactor.

**Figure 7 molecules-24-01532-f007:**
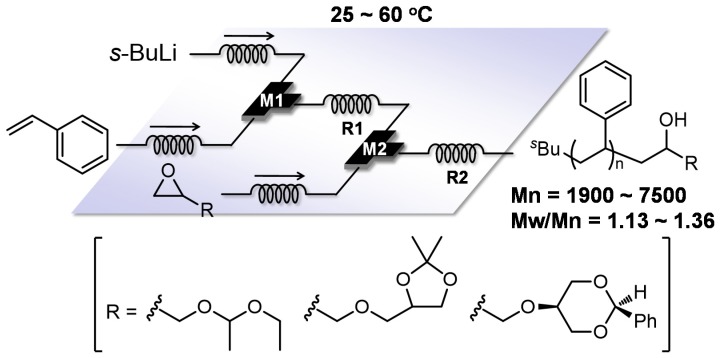
A flow microreactor system for anionic polymerization of styrene in THF initiated by *s*-BuLi and subsequent functionalization reaction with epoxides. M1, M2: micromixer; R1, R2: microtube reactor.

**Figure 8 molecules-24-01532-f008:**
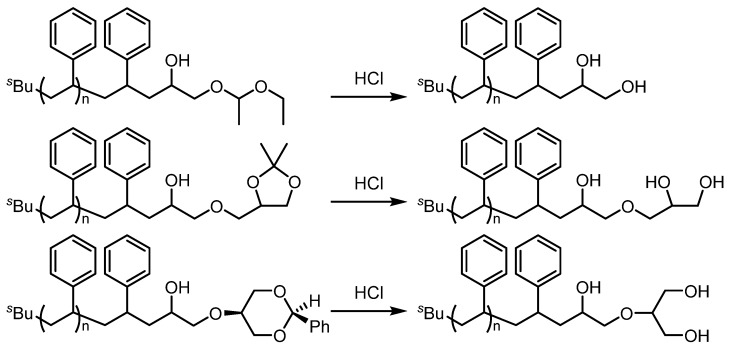
Synthesis of multi-hydroxyl end-functionalized polystyrenes.

**Figure 9 molecules-24-01532-f009:**
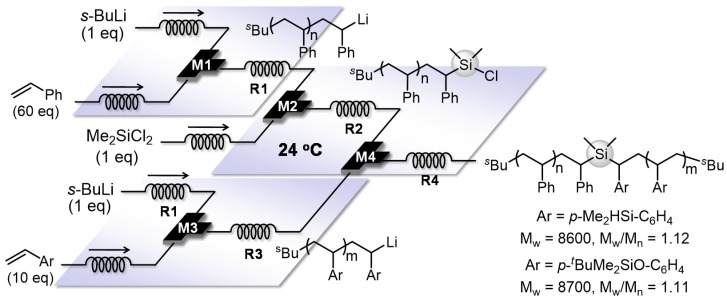
An integrated flow microreactor system for the synthesis of block copolymers having two different polymer chains on a silicon core (M1, M2, M3, M4: T-shaped micromixers; R1, R2, R3, R4: microtube reactors).

**Figure 10 molecules-24-01532-f010:**
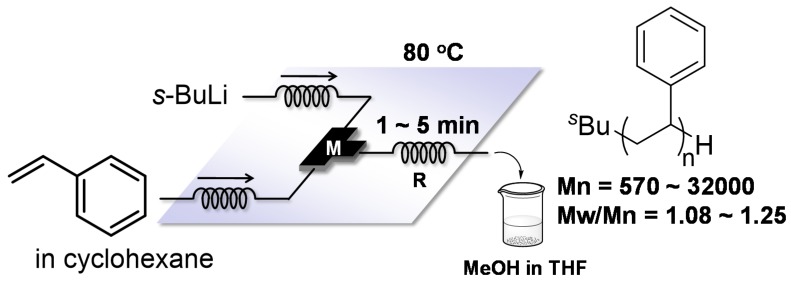
A flow microreactor system for anionic polymerization of styrene in cyclohexane at 80 °C initiated by *s*-BuLi. M: T-shaped micromixer; R: microtube reactor.

**Figure 11 molecules-24-01532-f011:**
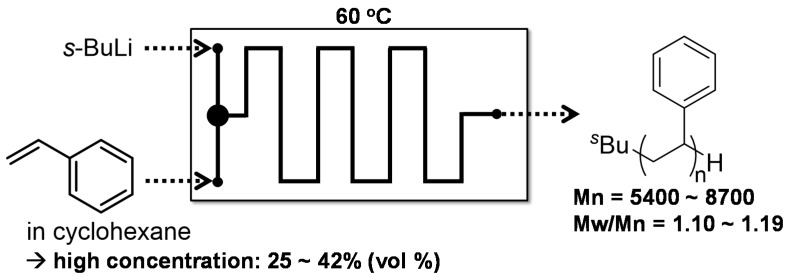
An aluminum-polyimide microfluidic device for anionic polymerization of styrene initiated by *s*-BuLi in cyclohexane at high concentrations at 60 °C.

**Figure 12 molecules-24-01532-f012:**
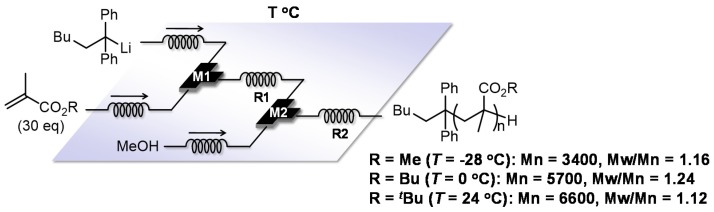
A flow microreactor system for anionic polymerization of alkyl methacrylates initiated by 1,1-diphenylhexyllithium. M1, M2: T-shaped micromixer; R1, R2: microtube reactor.

**Figure 13 molecules-24-01532-f013:**
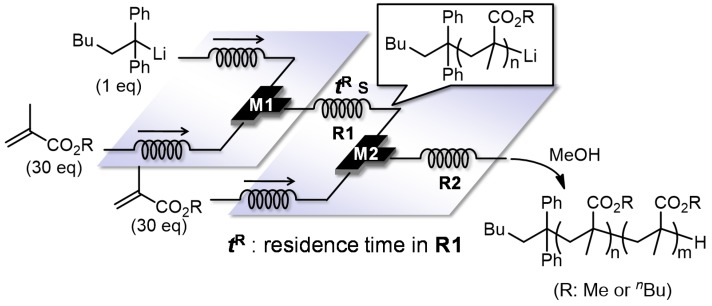
An integrated flow microreactor system for the sequential anionic polymerization of alkyl methacrylates initiated by 1,1-diphenylhexyllithium. M1, M2: T-shaped micromixer; R1, R2: microtube reactor.

**Figure 14 molecules-24-01532-f014:**
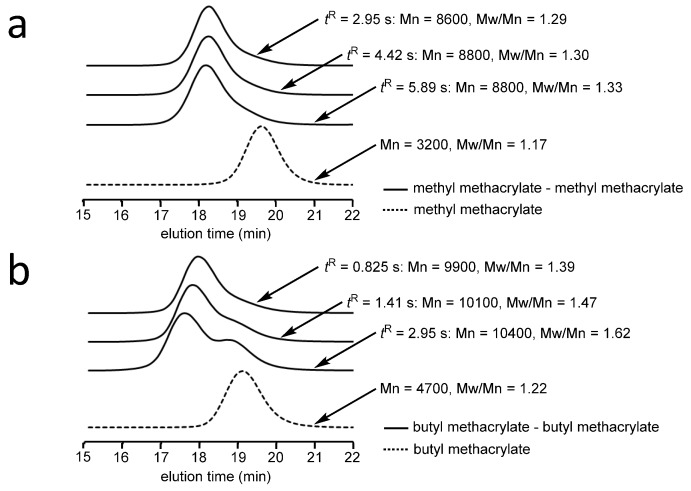
Size exclusion chromatography traces of polymers obtained in the integrated flow microreactor system. Effect of residence time on the molecular weight distribution. (**a**) methyl methacrylate–methyl methacrylate, (**b**) butyl methacrylate–butyl methacrylate.

**Figure 15 molecules-24-01532-f015:**
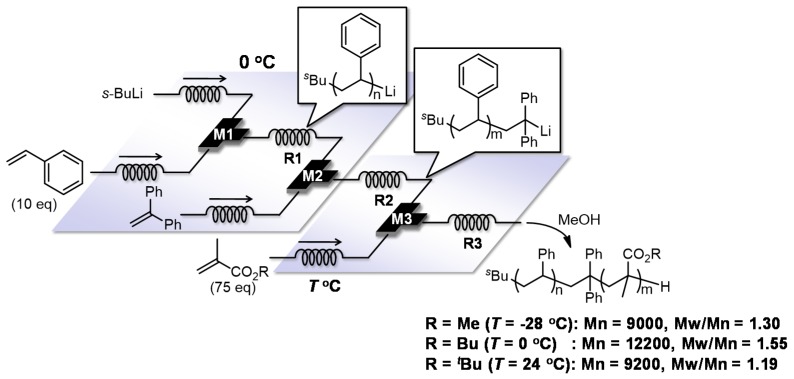
Integrated flow microreactor system for the anionic block copolymerization of styrene and alkyl methacrylates initiated by *s*-BuLi. M1, M2, M3: T-shaped micromixer; R1, R2, R3: microtube reactor.

**Figure 16 molecules-24-01532-f016:**
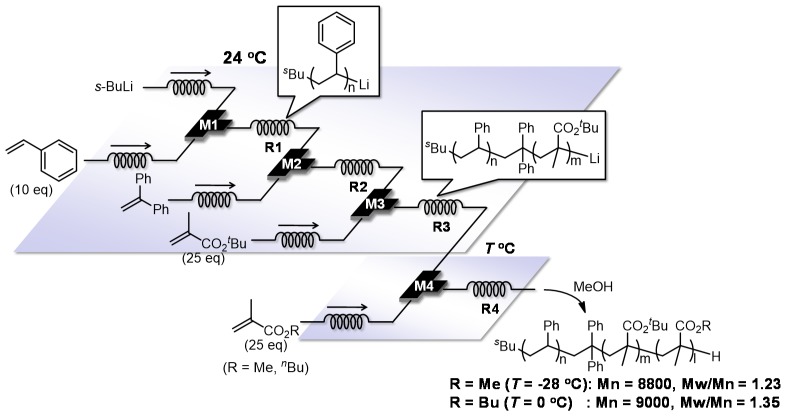
Integrated flow microreactor system for triblock copolymerization of styrene-*tert*-butyl methacrylate-alkyl methacrylate. M1, M2, M3, M4: T-shaped micromixer; R1, R2, R3, R4: microtube reactor.

**Figure 17 molecules-24-01532-f017:**
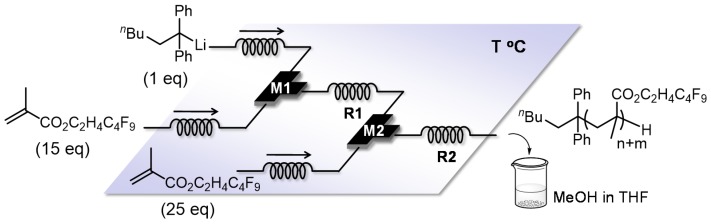
Flow microreactor system for the sequential polymerization of 2-(nonafluorobutyl)ethyl methacrylate (NFBEMA) initiated by 1,1-diphenylhexyllithium. M1, M2: T-shaped micromixer; R1, R2: microtube reactor.

**Figure 18 molecules-24-01532-f018:**
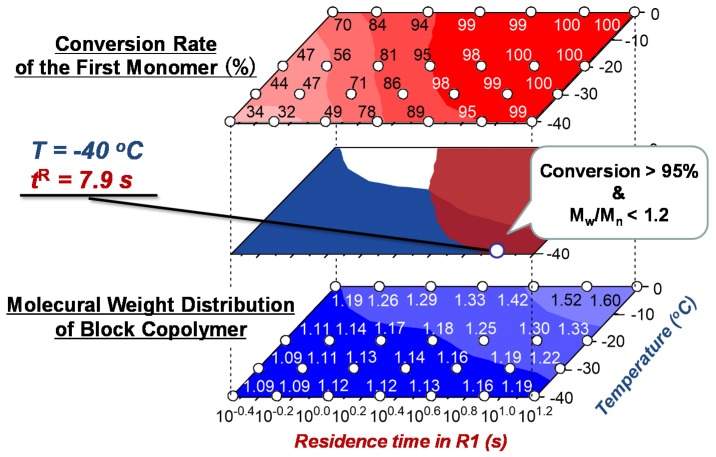
Three-dimensional temperature-residence time map of sequential polymerization of 2-(nonafluorobutyl)ethyl methacrylate (NFBEMA).

**Figure 19 molecules-24-01532-f019:**
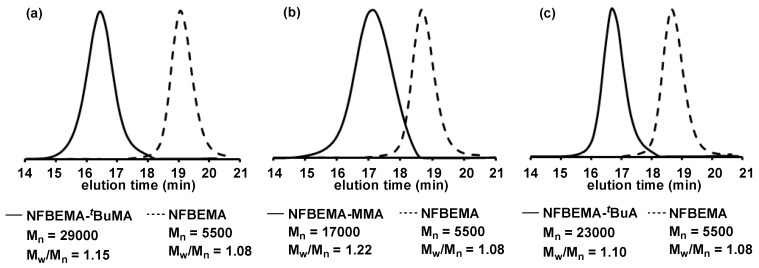
Size exclusion chromatography traces of block copolymers synthesized by the integrated flow microreactor system. Dashed line: 2-(nonafluorobutyl)ethyl methacrylate (NFBEMA) homopolymer. (**a**) 2-(nonafluorobutyl)ethyl methacrylate (NFBEMA) (polymerization temperature; T = −40 °C)/*tert*-butyl methacrylate (*^t^*BuMA) (T = 20 °C), (**b**) 2-(nonafluorobutyl)ethyl methacrylate (NFBEMA) (T = −40 °C)/methyl methacrylate (MMA) (T = −30 °C), (**c**) 2-(nonafluorobutyl)ethyl methacrylate (NFBEMA) (T = −40 °C)*/tert*-butyl acrylate (*^t^*BuA) (T = −20 °C).

**Figure 20 molecules-24-01532-f020:**
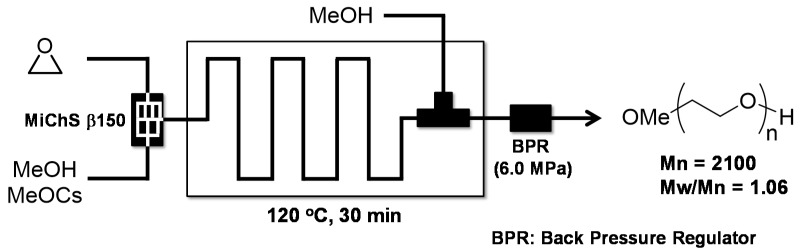
The flow microreactor system for the anionic ring opening polymerization of ethylene oxide.

**Figure 21 molecules-24-01532-f021:**
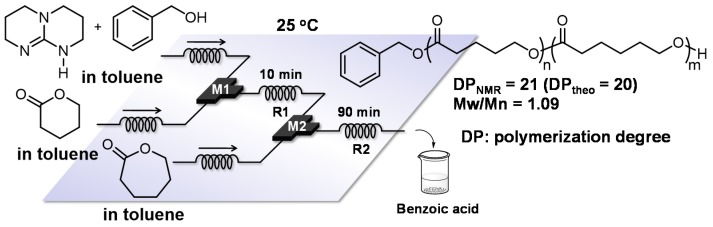
The integrated flow microreactor system for the block copolymerization of **ε**-caprolactone (CL) and **ε**-valerolactone (VL).

**Figure 22 molecules-24-01532-f022:**
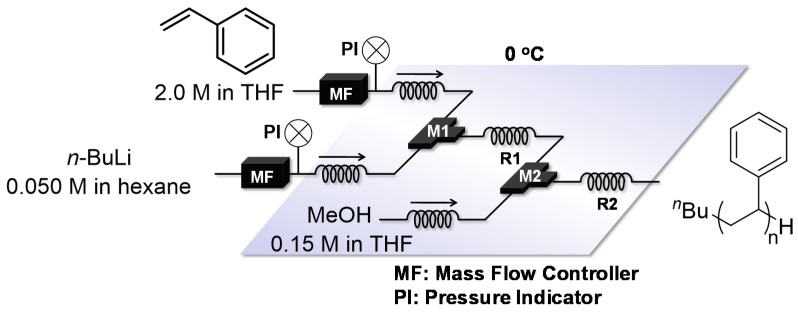
Schematic diagram of the flow microreactor system.

**Table 1 molecules-24-01532-t001:** Block copolymerization of alkyl methacrylates initiated by 1,1-diphenylhexyllithium using the integrated flow microreactor system.

Monomer-1	Monomer-2	Mn	Mw/Mn
MMA	-	3200	1.17
MMA	MMA	8600	1.29
BuMA	-	4700	1.22
BuMA	BuMA	9900	1.39
BuMA	*^t^*BuMA	9000	1.31
*^t^*BuMA	-	5300	1.13
*^t^*BuMA	*^t^*BuMA	10000	1.13
*^t^*BuMA	BuMA	9500	1.16
*^t^*BuMA	MMA	8400	1.15

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
