# Peer review of "Anionic Polymerization Using Flow Microreactors"

_molecules, 2019, doi:10.3390/molecules24081532_

Round 1

Reviewer 1 Report

Novel developments on the anionic polymerization employing flow microreactors are reported in this review. It is an excellent work showing the unique features of this methodology for the synthesis of well-defined polymers by anionic polymerization. It is a new area of research opening new opportunities even for the industrial application of this polymerization technique. Definitely, I propose the publication of this important manuscript after the authors take into account the following suggestions:

·         It is evident that rather low molecular weights are obtained from this polymerization technique. The authors should comment on the maximum molecular weight which can be achieved by this technique and compare with the conventional anionic polymerization.

·         It seems that the molecular weight distributions are slightly broader than the conventional anionic polymerization. Why does this happen?

·         Is it possible to polymerize dienes using these flow microreactors?

·         Is there any influence on the tacticity of polymethacrylates with this polymerization technique?

·         The extended purification of the reagents (monomers, solvents, etc.) is crucial for anionic polymerization. Is this purification procedure of equal importance for the discussed technique of polymerization?

·         The authors should add more detailed comments on the synthesis of end-functionalized polymers (degree of functionalization) and block copolymers (purity of the blocks). In certain cases SEC traces should be added.

·         Is it possible to influence the reactivity ratios during the synthesis of statistical copolymers employing this polymerization technique?

·         How easy is to synthesize complex macromolecular architectures, such as miktoarm stars, star block copolymers, block graft copolymers etc with this polymerization technique?

Author Response

(1)   It is evident that rather low molecular weights are obtained from this polymerization technique. The authors should comment on the maximum molecular weight which can be achieved by this technique and compare with the conventional anionic polymerization.

We added following sentences to comments on high molecular weights in the manuscript (page 12, line 336).

The polymers of various molecular weights (Mn = 5000 to Mn = 14000) with narrow molecular weight distribution (Mw/Mn = 1.08 - 1.16) were successfully synthesized simply by changing the relative flow rate and the concentration of solutions of styrene and initiator.

The higher molecular weight polymer synthesis using flow microreactors would be problematic from the viewpoint of pressure increase. Therefore, following comments were added in the manuscript (page 12, line 339).

In order to produce polymers with higher molecular weights, the suppression of pressure increase and/or the use of high pressure pump would be necessary.

Moreover, we have submitted a paper on the higher molecular weight polymer synthesis using flow microreactors, although it has not been accepted yet. We believe that such results will be reported in the near future.

(2)   It seems that the molecular weight distributions are slightly broader than the conventional anionic polymerization. Why does this happen?

In conventional batch reactors, it is well known that the purity of solvent and monomer become important to control molecular weight distributions. Even in flow microreactor polymerization, such points must be taken into consideration. Further purification of the reagents such as monomers and solvents would lead to the synthesis of narrower molecular weight distribution polymers. In addition, effective initiation by faster micromixing would give the formation of narrower molecular weight distribution polymers. Therefore, we added following sentences in the manuscript (page 3, line 107).

Polymer synthesis of narrower molecular weight distribution would be achieved by further purification of reagents and/or their effective mixing, although no such results are reported.

(3)   Is it possible to polymerize dienes using these flow microreactors?

Since the polymerization of dienes is industrially useful, it would be a very important research theme. We think that effective polymerizations of dienes together with vinyl monomers could be also achieved using a flow microreactor. However, to the best of our knowledge, such results have not been reported. Therefore, the details cannot be described in our present review.

(4)   Is there any influence on the tacticity of polymethacrylates with this polymerization technique?

To the best of our knowledge, such results on the tacticity of polymethacrylates have not been reported. However, the tacticity of polymethacrylates might be influenced using flow microreactors because optimized reaction temperatures between batch method and flow method would be different. Therefore, we added the following sentence in the manuscript (page 7, line 219).

Moreover, polymethacrylates synthesized by flow microreactor polymerization might give a different structure compared with that of the conventional batch method because optimized reaction temperatures would be different.

(5)   The extended purification of the reagents (monomers, solvents, etc.) is crucial for anionic polymerization. Is this purification procedure of equal importance for the discussed technique of polymerization?

As it was stated in above comment-(2), the purity of solvent and monomer must be taken into consideration with flow microreactor methods as well as batch methods. In fact, all the reactions shown in this review have been carried out using dry solvents and distilled monomers.

(6)   The authors should add more detailed comments on the synthesis of end-functionalized polymers (degree of functionalization) and block copolymers (purity of the blocks). In certain cases SEC traces should be added.

In the synthesis of complex macromolecular architectures, its efficiency is very important. However, the space in the manuscript is limited. Therefore, SEC trace in the block copolymerization of 2-(nonafluorobutyl) ethyl methacrylate (NFBEMA) with tert-butyl methacrylate, methyl methacrylate and tert-butyl acrylate were added as figure 19.

(7)   Is it possible to influence the reactivity ratios during the synthesis of statistical copolymers employing this polymerization technique?

The reactivity ratio of monomers depends on reaction temperatures. Therefore, because of higher temperatures compared to conventional batch reactors, we think that the reactivity ratio of monomers would be influenced by using flow microreactors. However, to the best of our knowledge, such results have not been reported. Therefore, such influences in details cannot be described in our present review.

(8)   How easy is to synthesize complex macromolecular architectures, such as miktoarm stars, star block copolymers, block graft copolymers etc with this polymerization technique?

We added following sentences in the manuscript (page 6, line 177).

Based on the present method, more complex macromolecular structures such as miktoarm stars, star block copolymers, and block graft copolymers would be also effectively synthesized.

Reviewer 2 Report

Dear Editor

The manuscript (review) “Anionic Polymerization Using Flow Microreactors” presented by Nagaki and Takahashi is really of interest and highlights some new achievements in field for anionic polymer synthesis assisted by flow chemistry. As pointed out by the authors this review is not the first but presents different points of view and deserves for publication in Molecules as it is.

I would like to just recommend for the authors to highlight a little bit better how flow chem can solve problems in anionic polymerizations and how the safety delivered by this technology is also an important point in this case. The authors could also address some weaknesses of the technology in terms of the solubility of the produced polymers, since clogging is a very dramatical problem to be solved. Overall, it is a very beautiful contribution to the literature of this fascinating enabling technology.

Author Response

(1)   I would like to just recommend for the authors to highlight a little bit better how flow chem can solve problems in anionic polymerizations and how the safety delivered by this technology is also an important point in this case. The authors could also address some weaknesses of the technology in terms of the solubility of the produced polymers, since clogging is a very dramatical problem to be solved. Overall, it is a very beautiful contribution to the literature of this fascinating enabling technology.

We also think that the use of flow microreactor system would be quite effective for the viewpoint of the improvement of some safety due to smaller reaction volume and higher surface to volume ratio compared to batch reactors. Therefore, we added following sentences in the manuscript (page 1, line 27).

Moreover, the use of flow microreactor system is quite effective for the viewpoint of the improvement of some safety due to smaller reaction volume and higher surface to volume ratio compared to batch reactors.

Reviewer 3 Report

The manuscript by Takahashi and Nagaki summarizes the field of anionic polymerization processes carried out in continuous-flow reactors.

Overall the review is well organized and the figures/schemes convey the intended information in a concise and clear manner. In this respect I have to note, however, that the topic, the arrangement, and many examples reported in the manuscript under exam are essentially identical to those found in two previous review chapters by the present corresponding author, in volumes of the 'Advances in Polymer Science' and, especially, 'Topics in Organometallic Chemistry' series [see refs. a,b below].

In general, the references discussed in the manuscript appear to adequately cover the literature in the field. However, in the event the contribution becomes accepted, some improvements in their handling are necessary. For instance, the group of papers cited in the sentence: 'Syntheses of complex polymers can be performed by integrating several flow microreactors[96-101].' (lines 134-135 in the manuscript) appear to be nearly out-of-context (they describe examples of author's work on continuous-flow transformations that are not polymerization processes); at the same time, no specific reference can be found to the paper dealing with the chemistry depicted in Fig. 6.

Other necessary changes/checks are included below for author' consideration.

- Fig. 6; please check whether the R group includes the aromatic fragment (C6H4) or not.

- Fig. 7; the main text mentions 'trans-2-phenyl-1,3-dioxane glycidyl ether (PDGE)', but the stereoisomer shown in the figure (rightmost R-group structure) turns out to be the cis one.

- Fig. 8; the wedge bond in the product of the bottom reaction scheme is not necessary.

- Fig. 17; contrarily to what it is stated in the caption, no styrene polymerization is shown in the picture. In my understanding, this figure, as well as Fig. 18, are related to the preliminary optimization step in the polymerization of NFBEMA; in the event, they could be better discussed in sect. 3.3 instead of sect. 3.4.

- inspect the text for minor English inconsistencies, mostly unusual choice of terms like 'the polymerization then consumes much longer time (line 89),'Smooth flow pomp...'  (line 325), etc.

[a] A. Nagaki, J-I Yoshida 'Controlled Polymerization and Polymeric Structures' (A. Abe, K. Lee, L. Leibler, S. Kobayashi Eds.) in 'Advances in Polymer Science', Springer (2014), ISBN 3319029193, 9783319029191
[b] A. Nagaki, J-I Yoshida, 'Organometallic Flow Chemistry' (T. Noël Ed.) in 'Topics in Organometallic Chemistry', Springer (2016), ISBN 3319332430, 9783319332437

Author Response

(1)   In general, the references discussed in the manuscript appear to adequately cover the literature in the field. However, in the event the contribution becomes accepted, some improvements in their handling are necessary. For instance, the group of papers cited in the sentence: 'Syntheses of complex polymers can be performed by integrating several flow microreactors[96-101].' (lines 134-135 in the manuscript) appear to be nearly out-of-context (they describe examples of author's work on continuous-flow transformations that are not polymerization processes); at the same time, no specific reference can be found to the paper dealing with the chemistry depicted in Fig. 6.

According to reviewer’s comment, we removed the references (refs-96 to 101). Moreover, references after ref-101 was renumbered.

Reference [92] corresponding to figure 6 was also added again (in page 4, line 139).

(2)   Fig. 6; please check whether the R group includes the aromatic fragment (C6H4) or not.

The R group in figure 6 doesn’t includes the aromatic fragment. Therefore, figure 6 was revised.

(3)   Fig. 7; the main text mentions 'trans-2-phenyl-1,3-dioxane glycidyl ether (PDGE)', but the stereoisomer shown in the figure (rightmost R-group structure) turns out to be the cis one.

The original paper shows same description and figure. In order to avoid misunderstandings, we removed the word of “trans” (in page 5, line 151).

(4)   Fig. 8; the wedge bond in the product of the bottom reaction scheme is not necessary.

Figure 8 was revised.

(5)   Fig. 17; contrarily to what it is stated in the caption, no styrene polymerization is shown in the picture. In my understanding, this figure, as well as Fig. 18, are related to the preliminary optimization step in the polymerization of NFBEMA; in the event, they could be better discussed in sect. 3.3 instead of sect. 3.4.

The caption of figure 17 was revised as follow.

Figure 17. Flow microreactor system for the sequential polymerization of 2-(nonafluorobutyl)ethyl methacrylate (NFBEMA) initiated by 1,1-diphenylhexyllithium. M1, M2: T-shaped micromixer; R1, R2: microtube reactor.

(6)   inspect the text for minor English inconsistencies, mostly unusual choice of terms like 'the polymerization then consumes much longer time (line 89),'Smooth flow pomp...'  (line 325), etc.

We revised two sentences as follow (page 3, line 87 and page 12, line 342).

On the other hand, with nonpolar solvents, the polymerization can be carried out at higher temperatures, but much longer reaction time is needed for completion.

The Smoothflow pump has a structure which allows a solution to be contained in its pump head during a resting time. This feature offers significant superiority for a reaction operation with a reactant solution which is unstable to air and/or water.